# Human SMILE-Derived Stromal Lenticule Scaffold for Regenerative Therapy: Review and Perspectives

**DOI:** 10.3390/ijms23147967

**Published:** 2022-07-19

**Authors:** Mithun Santra, Yu-Chi Liu, Vishal Jhanji, Gary Hin-Fai Yam

**Affiliations:** 1Corneal Regeneration Laboratory, Department of Ophthalmology, University of Pittsburgh School of Medicine, Pittsburgh, PA 15213, USA; mithun.santra@pitt.edu (M.S.); jhanjiv@pitt.edu (V.J.); 2Tissue Engineering and Cell Therapy Group, Singapore Eye Research Institute, Singapore 169856, Singapore; liuchiy@gmail.com; 3Ophthalmology and Visual Sciences Academic Clinical Program, Duke-NUS Medical School, Singapore 169857, Singapore; 4McGowan Institute for Regenerative Medicine, University of Pittsburgh, Pittsburgh, PA 15213, USA

**Keywords:** stromal lenticules, corneal regeneration, extracellular matrix scaffold, tissue engineering

## Abstract

A transparent cornea is paramount for vision. Corneal opacity is one of the leading causes of blindness. Although conventional corneal transplantation has been successful in recovering patients’ vision, the outcomes are challenged by a global lack of donor tissue availability. Bioengineered corneal tissues are gaining momentum as a new source for corneal wound healing and scar management. Extracellular matrix (ECM)-scaffold-based engineering offers a new perspective on corneal regenerative medicine. Ultrathin stromal laminar tissues obtained from lenticule-based refractive correction procedures, such as SMall Incision Lenticule Extraction (SMILE), are an accessible and novel source of collagen-rich ECM scaffolds with high mechanical strength, biocompatibility, and transparency. After customization (including decellularization), these lenticules can serve as an acellular scaffold niche to repopulate cells, including stromal keratocytes and stem cells, with functional phenotypes. The intrastromal transplantation of these cell/tissue composites can regenerate native-like corneal stromal tissue and restore corneal transparency. This review highlights the current status of ECM-scaffold-based engineering with cells, along with the development of drug and growth factor delivery systems, and elucidates the potential uses of stromal lenticule scaffolds in regenerative therapeutics.

## 1. Introduction

### 1.1. Human Cornea Anatomy, Composition and Cell Types

A healthy cornea with high transparency and optimal refractivity is paramount for vision. As the first layer of the eye, the cornea refracts light onto the lens, which then reaches the retina. It also provides mechanical support and protection to the intraocular tissues and defense against pathogens. The adult cornea measures 11 to 12.5 mm in diameter, with a mean anterior corneal curvature radius of around 8 mm. The corneal thickness is 500 to 600 μm (an average of 540 μm in the center and 700 μm in the periphery) with a refractive index of 1.38 [1,2]. The human cornea comprises five layers: the epithelium, Bowman’s layer, stroma, Descemet’s membrane, and innermost corneal endothelium (Figure 1).

The corneal epithelium is a non-keratinized stratified squamous epithelium. It functions as a physical barrier against external hazards and protects the eye from chemicals and microbes, which helps in maintaining visual acuity [3]. It is the only corneal tissue that undergoes both maintenance and injury-triggered regeneration due to the presence of the epithelial stem cell population in the peripheral limbus. The Bowman’s layer is a collagen-rich basement membrane of the corneal epithelium [4]. Corneal stroma, the thickest layer of the cornea, accounts for 80–90% of the overall corneal volume [5]. It provides mechanical strength to the cornea, comprising highly structured collagen fibers and a stromal matrix. A syncytial network of corneal stromal keratocytes populates the inside of the stroma. These cells are quiescent and are indispensable for stromal homeostasis, as they primarily produce the collagens and proteoglycans that comprise the stromal extracellular matrix (ECM) [6]. Parallel layers of collagen fibrils (predominantly type I and V) form an organized matrix architecture with an orthogonally aligned pattern for undisturbed light passage without scattering. Furthermore, small leucine-rich keratan sulfate proteoglycans (SLRPs) (such as lumican, keratocan, mimecan, and decorin) regulate the stromal collagen fibrillar spacing and play a significant role in maintaining the structural integrity of the stromal matrix [7]. Descemet’s membrane is the basement membrane of the innermost corneal endothelium. It is composed of collagen types IV and VIII [8,9]. The corneal endothelium is lined with a monolayer of tightly packed hexagonal-shaped corneal endothelial cells. Its continuous pump/leak activity regulates stromal hydration preventing corneal edema and loss of transparency [10].

### 1.2. Corneal Blindness and Conventional Treatments with Tissue Grafting

The cornea is susceptible to abrasive insults, including mechanical, chemical, and thermal injury and infections. Globally, corneal blindness ranks fifth after refractive errors, cataracts, age-related macular degeneration, and glaucoma. In a recent report issued by the World Health Organization (WHO), around 2.2 billion people have vision impairment, including 4.2 million with unaddressed corneal opacities (WHO *World report on vision*, October 2019; https://www.who.int/publications/i/item/9789241516570, accessed on 13 February 2022). Corneal transplantation (keratoplasty) is the primary and most successful treatment modality for severe corneal opacities (Figure 2). However, the scarcity of transplantable donor tissues has limited the treatment outcomes. A recent report projected that 12.7 million people worldwide are waiting for corneal transplantations [11,12]. The highest corneal transplantation rate is in the USA (199 cases per million), followed by Lebanon (122 per million) and Canada (117 per million). However, the transplantation rate for the other 116 countries is only 19 per million. Approximately 53% of the world’s population has no access to corneal transplantation, and only 1 out of 70 patients with corneal blindness can access a transplantable donor cornea [11].

Although corneal transplantation is the most frequent and successful type of organ transplant worldwide, various postoperative complications have been reported, such as astigmatism, infection, wound dehiscence, and graft rejection [13,14]. Astigmatism affects almost 15–31% of patients (greater than five diopters) undergoing penetrating keratoplasty [15].

### 1.3. Corneal Wound Healing and Scar Development

Corneal opacity or scarring results from injury, infection, or hereditary or inflammatory corneal diseases. After injury, the corneal epithelium, stroma, and endothelium heal with different but interlinked mechanisms [16]. Corneal epithelial cells and stromal keratocytes produce and release cytokines to initiate epithelial–stromal interaction for coordinated wound healing. IL-1α and platelet-derived growth factor (PDGF) secreted by the injured epithelium induce stromal keratocytes to be activated and undergo fibrosis (more details in the following paragraph) [4,16,17]. Moreover, keratocyte-secreted Growth Factors (GF), such as keratinocyte growth factor (KGF) and hepatocyte growth factor (HGF), aid epithelial wound healing by influencing cell behaviors, including cell migration and proliferation [16,18]. Adult limbal epithelial stem cells are also activated and generate transit-amplifying cells that migrate and regenerate the corneal epithelium [19,20]. On the other hand, the corneal endothelium has a specific pump and leak action to keep the corneal stroma in a partially dehydrated status for corneal clarity. Endothelial wounds mostly heal through cell shape enlargement and the sliding of adjacent endothelial cells rather than mitosis, as adult corneal endothelial cells are post-mitotic quiescent and lack regenerative capacity. Hence, endothelial cell loss, if severe, causes the cell density to drop below the threshold needed to maintain an efficient pump/leak action, resulting in corneal edema and opacities [21,22].

Inside the corneal stroma, the dominant population of stromal keratocytes regulates stromal homeostasis by producing and depositing stromal-specific collagens and matrix proteoglycans, and this maintains the structural integrity with minimal light scattering. Our recent review detailed the role of stromal keratocytes in corneal health and visual functions [17]. Mature keratocytes are quiescent and show minimal mitosis in adult life. There is an estimated 0.45% cell loss per annum. Keratocyte-secreted maspin inhibits cell migration and stimulates cell adhesion to ECM [23]. Keratocytes also have phagocytic functions [17,24]. Upon injury, keratocytes at the wound site undergo apoptosis due to IL-1α and PDGF produced by the corneal epithelium. Cells in the peripheral regions are activated to re-enter the cell cycle. This event generates proliferative and motile stromal fibroblasts (SFs) with an accompanying actin cytoskeleton-mediated morphological change from a dendritic to a spindle shape. The activated SFs have a loss of keratocyte phenotypes, including the expression of keratan sulfate proteoglycans and stromal crystallins. On the other hand, there is an increased production of ECM proteins, including collagen I, fibronectin, and biglycan, an initiation of fibronectin receptor and integrin (α5β1) interaction, and an expression of IL-1α-controlled metalloproteinases (MMPs). These features are related to stromal tissue remodeling that alters the stromal matrix architecture, resulting in corneal haze development [25]. Further generation of contractile myofibroblasts due to the presence of pro-fibrotic transforming growth factor (TGF)-β isoforms exaggerates these fibrotic events with disorganized ECM and excessive matrix contraction, forming scars and opacities. SFs also produce growth factors (GFs) to trigger neovascularization into the avascular stroma and the expression of MMP to degrade stromal collagens to assist new blood vessel formation and penetration [26].

## 2. Corneal Regenerative Approach: Cell-Based vs. Scaffold-Based Strategies

### 2.1. Cell-Based Therapies

Conservative treatments (topical anti-inflammatory medications or steroid eye drops and minor surgeries) and donor tissue grafting are the major options for managing corneal opacities and scarring. The limited supply of transplantable donor tissue and the risk of allogenic graft rejections have urged researchers to look for alternative options to restore corneal functions. The regenerative ability of corneal cells has led to extensive research into regenerating and reconstructing various layers of the cornea, starting from the corneal epithelium [27,28,29,30] to current developments in the corneal stroma [31,32,33] and endothelium [21,34,35,36,37]. In the regeneration of corneal layers, cell and tissue engineering have emerged as essential strategies for developing novel substitutes [38]. Specific corneal stem cells have immense potential to induce respective tissue regeneration through multiple modes of action, including: (1) differentiation of limbal epithelial stem cells to corneal epithelial cells and corneal stromal stem cells (CSSCs) into keratocytes to replenish the lost/damaged cells; (2) activation of resident cells to assist in tissue repair; (3) secretion of regenerative molecules to reduce tissue inflammation, promote tissue remodeling, and activate the signaling pathways associated with tissue healing and regeneration [31]. Different studies have reported the differentiation of CSSCs, extraocular mesenchymal stem cells (MSCs), and induced pluripotent stem cells (iPSCs) to survive and differentiate in vivo along the keratocyte lineage and synthesize new collagens in the recipient stroma [31,38]. MSC treatment also reduces pre-existing scars and stromal defects, resulting in improved corneal clarity [38]. In addition, the immunomodulatory properties of MSC make them ideal for corneal regeneration in syngeneic, allogeneic, and even xenogeneic scenarios [39,40].

Despite the potential of cell-based therapies, there are limitations regarding the direct delivery of cells. The therapeutic efficacy depends on the precise location and retention of viable cells at the target site and their functions in the new niche [41]. However, a number of attempts have not been met with critical success due to post-transplantation complications, such as poor cell localization, short-term survival, and side effects (including the transition to other cell types and tumorigenesis) [41]. Without appropriate data from corneal research, we have found from other studies that less than 5% of transplanted cells reach the target site after intravenous administration, and their survival rate can be as low as 1% [42,43,44,45].

### 2.2. Scaffold-Based Cell Delivery Strategies

Scaffold-based cell delivery has the potential to overcome some limitations associated with cell-based therapy, such as poor cell localization, as mentioned above. Bioscaffolding can assist in efficient cell delivery; at the same time, cells introduced inside the scaffold can create 3D tissue analogs. It also provides a suitable matrix environment for cell adhesion and growth [46] and maintains cellular functions, such as cytoskeleton reorganization, integrin activation, gene expression, and ECM organization [47]. Compared with cell suspensions, cell-laden scaffolds show higher cell viability and better integration at the host site [48]. Furthermore, in tissues where cell orientations are necessary, a good alignment of cells and ECM is feasible. Such cell guidance inside the scaffold can assist host tissue regeneration and functional recovery [47]. In tissues and organs, such as corneal stroma, tendons, bones, and skeletal muscles, specific cell–ECM alignment is crucial for organ function [49]. In the corneal stroma, keratocytes are sparsely located between the parallel running collagen lamellae, and this is essential to minimize incident light scattering for perfect light transmission [17,50]. In tendons, the presence of a unique cell–matrix alignment provides substantial resistance and exceptional mechanical properties to the tissue in that axis [51,52]. Other similar examples can be observed in cartilage [53], dental enamel [54], and basal epithelium [55]. Reproducing those cell–matrix alignment patterns within the tissue-engineered substitutes can generate a more native physiological representation of biological tissues, leading to a better recovery of tissue functions.

Engineering cells in a scaffold also warrant mechanical support and protection for the cells [37]. Last but not least, a bioscaffold used for corneal tissue engineering should have excellent optical characteristics, biocompatibility, and stability [56].

The ECM is a non-cellular component present in every tissue and organ in our body, providing the physical scaffolding support and essential biochemical, biophysical, and biomechanical signals necessary for tissue morphogenesis, differentiation, and homeostasis. Therefore, an ECM-derived scaffold is an ideal biocompatible material for cell incorporation compared with other biosynthetic platforms. In corneas, stromal matrix scaffolds have shown the potential to be used for such a purpose due to the good preservation of the native ECM structure [57].

#### Decellularization of ECM Tissues for Scaffold-Based Engineering

Decellularization aims to eliminate cellular and antigenic molecules, including genetic materials while preserving the structural, biochemical, and biomechanical properties of the matrix scaffold [46,58,59]. In the context of tissue engineering, a decellularized scaffold provides a native-like ECM environment with high bioactivity and compatibility for cell–ECM interaction, and this promotes subsequent cell adhesion, proliferation, and survival [60,61]. In addition, it offers the advantage of a remarkable similarity with the tissue to be replaced. After delivery to the target site, the decellularized ECM can be repopulated with the recipient cells to produce an integrated cell–tissue composite. Various decellularization methods have been developed to efficiently remove immunogenic cellular materials, hence maintaining low immunogenicity [62]. A good decellularization protocol would produce non-immunogenic ECM with the original structural integrity and a preserved protein content, including collagen, fibronectin, and glycosaminoglycans (GAGs). As a criterion to be justified as non-immunogenic, decellularized ECM should contain less than 50 ng of double-stranded DNA per mg ECM dry weight, and the residual DNA fragments must be less than 200 bp in length [58,63].

In recent years, different protocols have been developed to decellularize human corneal tissue [64,65,66,67,68,69,70]. The different approaches, methodology details, reagent requirements, and efficiency, as well as limitations, have been broadly reviewed recently [63,71]. Ideally, decellularization techniques aim to completely remove cellular and antigenic materials. However, most protocols inevitably cause tissue disruption and a loss of intrinsic biological cues. Similarly, residues of decellularization reagents might be retained in the resulting matrix, which could have a negative influence on further engineering events and/or tissue transplantation. From a manufacturing point of view, simple protocols with fewer steps and minimal reagent use are desirable. Stringent checks on decellularized tissue architecture, cellular material, and protein content, as well as any residual reagents, are necessary.

## 3. Stromal Lenticules from SMILE

Small incision lenticule extraction (SMILE) is an FDA-approved laser refractive surgery that has become increasingly popular for the correction of myopia and astigmatism. Compared with excimer laser surgery (e.g., LASIK, laser-assisted in situ keratomileusis), SMILE is a “flap-less” surgery that involves the creation of an intrastromal lenticule and a peripheral arc-shaped incision using a femtosecond laser (FSL). The stromal lenticule is removed through the incision opening (Figure 3). The thickness of the lenticule created is determined by the refractive power to be corrected (approximately 15 μm in the center for one diopter spherical equivalent). For myopic treatment, the FSL-created lenticule is thicker in the center and thinner at the periphery, and its removal flattens the central cornea and achieves therapeutic effects. The lenticule is typically 6.0–6.5 mm in size and round in shape [72]. A meta-analysis of randomized controlled trials and comparative studies comparing SMILE and LASIK by Zhang et al. found no significant difference between them in many areas except corneal sensation and postoperative dry eye, which favored SMILE [73]. Given its flap-less and minimally invasive technique and its promising outcomes, the SMILE procedure will be an upcoming option for more patients undergoing refractive surgeries. Additional resources regarding SMILE can be referenced at https://www.aao.org/eye-health/treatments/small-incision-lenticule-extraction-smile (accessed on 28 June 2022).

As per a recent report by Schallhorn et al., more than 3.5 million SMILE procedures have been performed globally in the 10 years since its commencement in September 2011 (Schallhorn 2021 Cataract & Refractive Surgery Today https://crstoday.com/wp-content/uploads/sites/4/2021/02/0221CRST_F_Schallhorn.pdf, accessed on 2 July 2022). This accounts for an approximate mean of 350,000 procedures per year. Over 2000 surgeons are offering this refractive surgery in 70 countries. It is the most popular laser vision correction procedure in Korea and is on the path to being the dominant procedure in China. Furthermore, over 600 published peer-reviewed articles involve SMILE-related studies (https://crstoday.com/articles/feb-2021/smile-latest-and-limits/, accessed on 2 July 2022).

SMILE lenticules are neatly cut discs of native biomaterial that are ultrathin (about 30–140 μm thick, depending on the diopter correction) [74], transparent, avascular, and mechanically strong with a well-organized collagen-rich ECM composition, and are obtained from young, healthy corneas. Hence, discarding them is an enormous waste and a huge loss of opportunities for potential applications. With the constantly growing number of SMILE surgeries, these lenticules are a valuable resource deserving of extensive study so that they can be recycled or upcycled for therapeutic uses.

### 3.1. Utilization of SMILE-Derived Stromal Lenticules

SMILE-derived stromal lenticules have great potential to be used in multiple ways, for example, in tissue repair for wound healing or in the tissue addition process to improve tissue strength and integrity (such as in ectasia). Different studies have reported the use of stromal lenticules originating from SMILE in various areas of corneal surgeries. Lenticule implantation can be an effective treatment modality for hyperopia. The first SMILE lenticule implantation in a human subject was reported by Pradhan et al., where they implanted a −10 D lenticule into an FSL-created pocket of a patient with +11.25 D hyperopia, resulting in a decreased spherical equivalent and improved corneal topography [75]. Similar work was reported by Seiler et al. to correct hyperopia, and this new surgical procedure was termed lenticule intrastromal keratoplasty (LIKE) [76]. Different reports have described the LIKE procedure as a novel tissue-addition-based treatment for advanced keratoconus that improves patients’ vision. Doroodgar et al. showed that customized SMILE lenticule implantation improved corneal curvature and the refraction of keratoconic corneas [77]. Negative meniscus-shaped lenticular addition was also found to induce a flattening of the cone region while increasing corneal thickness [78].

In a recent case report, post-LASIK corneal ectasia was treated with the implantation of an allogeneic stromal lenticule, and a significant decrease in the steep keratometric values was achieved [79]. For the treatment of presbyopia, synthetic inlays have been designed to enhance patients’ near vision. Compared with synthetic inlays, SMILE-derived stromal lenticules are biological and biocompatible, having greater potential to integrate with host corneal tissue with less disturbance to the oxygen and nutrient flow. Therefore, corneal integrity would be maintained, and the risk of tissue necrosis after implantation would be minimized. Table 1 summarize the safety evaluation and treatment outcomes of using allogeneic and autologous SMILE lenticules.

Compared with full-thickness or lamellar keratoplasty, the risk of rejection for lenticule implantation is theoretically lower. This could be due to the following: (i) thin lenticules have a less antigenic load to elicit an immunological response, and they contain fewer stromal keratocytes, without epithelial or endothelial cells having more antigenic reactions; in particular, decellularized stromal lenticules have even lower immunogenicity and are a better option for allogeneic use; and (ii) after implantation into the stroma, the lenticule is not in direct contact with tear and aqueous humor, which contains triggering factors for immune responses [63,80,81].
ijms-23-07967-t001_Table 1Table 1Clinical studies of SMILE lenticule implantation.Corneal ConditionsLenticule TypesProcedureStudy SubjectsConclusionReferencesPresbyopiaAllogenic corneal inlay prepared from SMILE-derived lenticulesPresbyopic allogenic refractive lenticule (PEARL) inlay4 patients with emmetropia and presbyopia It demonstrated the safety and efficacy of a PEARL corneal inlay for presbyopic correction.[80,82]HyperopiaAutologous SMILE-derived lenticules Lenticule implantation5 patients with 1 eye myopic and the other hyperopicImplanting an autologous SMILE-derived lenticule for hyperopia correction was safe, effective, and stable.[83]Allogenic SMILE-derived lenticuleFemtosecond laser-assisted keyhole endokeratophakiaA 23-year-old aphakic patientTreatment corrected hyperopia to 50% of the intended correction.[75]KeratoconusAllogenic SMILE-derived lenticulesFemtosecond laser-assisted stromal lenticule implantation combined with accelerated collagen cross-linking6 patients with progressive keratoconusCombined lenticule implantation and collagen cross-linking is a feasible option to treat low to moderate keratoconus.[84]Corneal dystrophyAllogenic SMILE-derived lenticulesEpikeratophakia combined with photo-therapeutic keratectomy6 patients with recurrent corneal dystrophyA feasible treatment with improvement in vision and a good safety profile.[85]Micro-perforationsAllogenic SMILE-derived lenticule patch graftGlued lenticule patch graft transplantation7 eyes of 7 patientsA safe, feasible, and inexpensive surgical option. [86]


### 3.2. SMILE Lenticule Storage and Customization

Since the inception of SMILE surgery, several studies have explored the optimal storage conditions for lenticules to maintain their tissue viability, structure, and clarity. Liu et al. demonstrated that human lenticules stored in various reagents at 4 °C or room temperature for up to 48 h had similar tissue clarity and structural integrity [87]. This offers a convenient and feasible option for short-term lenticule transportation to more extensive facilities for long-term storage. Another study by Ganesh et al. demonstrated that the long-term cryopreservation of lenticules did not alter their structural integrity and was safe for use in allogeneic transplantation [88]. In the study, SMILE-derived lenticules cryopreserved for 178 days were transplanted to one aphakic and eight hyperopic eyes in patients, with no evidence of tissue rejection or loss of best-corrected visual acuity during the follow-up period (38–310 days).

SMILE-derived lenticules can be customized for the purpose of re-implantation. A recent study showed that lenticules were successfully thinned and reshaped using excimer laser ablation [89]. Controlled lenticular dehydration status was necessary for correct stromal thinning. Another study by Bandeira et al. revealed the density and excitatory response of neurites and Schwann cells (SCs) in fresh and cryopreserved SMILE-derived lenticules. Although the stromal neurites showed variations in density related to SMILE lenticular thickness and cryopreservation, these neurite residues could retain minimal functionality with the presence of SC support and an excitatory response, suggesting the potential advantage of re-innervation after lenticular implantation [90,91].

### 3.3. Decellularized SMILE Lenticules in Corneal Bioengineering

The use of decellularized SMILE scaffolds in corneal bioengineering is a novel practice, and its potential is yet to be explored. However, a handful of studies have been conducted to date showing the potential use of lenticule scaffolds in cell culture and delivery. One study evaluated the feasibility of an MSC culture on decellularized lenticules, supporting MSC differentiation into corneal epithelial cells [92]. Their findings offer the prospect of a novel therapeutic modality of SMILE-derived lenticules in regenerative corneal tissue engineering. Moreover, patient-derived induced pluripotent stem cells (iPSCs) seeded on decellularized lenticules were found to differentiate into corneal epithelial-like cells with the formation of a coherent stratified squamous epithelial sheet [93]. This could be an autologous corneal epithelial replacement for persistent corneal epithelial defect due to bilateral total limbal stem cell deficiency. Another study showed that decellularized lenticules significantly increased bleb survival and decreased intraocular pressure postoperatively in glaucoma filtration surgery on a rabbit model by acting as a physical adhesion barrier [94]. This result suggests that stromal lenticules could prevent postoperative conjunctiva–sclera adhesion and fibrosis, representing a novel anti-fibrotic management method for trabeculectomy.

### 3.4. Current Obstacles to the Use of SMILE Lenticules for Clinical Applications

The promise of lenticule implantation is exciting, but much remains to be conducted to bring this technique into mainstream practice.

No standard methods with high reproducibility are available to customize lenticules prior to implantation in patients that require specific settings. Although several studies have shown some degree of modification to have the appropriate thickness for refractive correction or to be mechanically strengthened for ectasia treatment or for the elimination of immune-prone biomolecules for allogenic lenticules by decellularization [68,89,91], standard methods that are widely acceptable to the clinical community are yet to be established.Lenticule storage and biobanking systems are still being developed in a few countries, such as Singapore (https://www.straitstimes.com/singapore/health/singapore-launches-first-bank-in-asia-for-eye-surgery-patients-to-freeze-piece-of, accessed on 3 July 2022) [87,95]. A structured regulatory and organizational framework for lenticule processing and banking is important for the successful operation, standardization, and quality assurance of lenticule products and for safe and effective treatments for patients.

The goal is to set standards and protocols for the lenticule implantation technique for the benefit of ophthalmologists and patients. Multicenter trials are also encouraged to generate knowledge and to avoid single-center bias. This will promote the useful recycling of the extracted lenticules instead of them being discarded. That equates to ~0.3 million or more discarded lenticules per year, which is a huge waste of native and healthy biological tissues.

## 4. Applications of ECM Scaffolds from Other Sources in Tissue Engineering

In the above sections, we have described the potential usages of SMILE-derived stromal lenticules, primarily through tissue implantation. So far, very few studies have reported other benefits, such as recellularization, drug delivery, and incorporation of growth factors (GF) or nanoparticles (NP). In order to gain a precise and practical perspective on the potential development of using SMILE lenticules in tissue engineering, we have summarized various major studies on the in vitro modifications of ECM scaffolds derived from other tissue sources and their therapeutic applications in pathologies. We anticipate that these valuable data will shed light on the exploration of the novel potential of SMILE-derived lenticules in future applications.

### 4.1. Recellularization on Stromal Scaffolds and Potential Applications

Decellularized ECM scaffolds can be repopulated with cells to generate viable and transplant-worthy tissues. For successful recellularization, an optimal cell seeding and physiologically relevant culture methodology first need to be established. Several reviews have provided details on different recellularization protocols and their benefits and drawbacks [96,97]. Earlier studies reported repopulating decellularized matrices using cell lines. Due to the cells being modified to grow perpetually, these products contain the risk of tumorigenesis or fibrogenesis; hence they have relatively low translational potential [98,99]. Thus, primary cultured cells are required to be used to safeguard the correct cell phenotypes relevant to the target tissues. In a study by González-Andrades et al. using sodium-chloride-decellularized porcine corneal tissues, serum-expanded primary human keratocytes were seeded on the tissue surface. After 14 days of culture, the ALDH1-expressing human cells migrated into the acellular stroma [100]. More recently, Alió del Barrio et al. reported the recellularization of adipose-derived MSCs on thin decellularized stromal laminas in vitro, followed by transplantation to six patients with advanced keratoconus [70]. In addition, the freeze-drying of decellularized porcine corneas was reported to induce tissue pore formation for cell penetration to deeper levels in vitro [101]. For the purpose of repopulating recipient tissues following scaffold transplantation, the methodology is to seed cells directly onto the surface of the decellularized scaffold. After implantation, the stromal repopulation relies on the capacity of the cells to migrate from the scaffold to the surrounding regions. A recent study transplanting labeled cell-loaded testis scaffolds showed appropriate cell migration to the recipient tissue, indicating that a testicular ECM scaffold could be a promising vehicle to support cell transplantation [102]. A decellularized heart tissue scaffold was also reported to successfully deliver exogenous cardiomyocytes to the retroperitoneum of recipient animals [103].

In vitro expanded cells under controlled culture conditions (following the guidelines of good manufacturing practices (GMP)) are seeded onto the scaffold, followed by the culture/maturation of the newly cellularized tissues and implantation into the recipient organs. In deep tissue organs, such as the heart or lungs, endogenous vasculatures need to be established to sustain the survival of transplanted tissues. This is because oxygen diffusion can reach up to a depth of 150–200 μm from the tissue surface [104]. However, the surface location of the cornea provides oxygen availability and nutrients from aqueous humor and tears. This allows the feasible use of cell-engineered ECM constructs for corneal reconstruction.

### 4.2. Tissue Regeneration with ECM Scaffolds

#### 4.2.1. Cell-Scaffold Interactions Guide Tissue Formation

During development and tissue repair, changes in ECM organization and composition convey information signals that influence various cellular events, including cell growth, differentiation, and phenotypes [105]. They can be delivered via mechanical signaling from the ECM to target cells through specific cell surface receptors, such as integrins. Various ECM proteins (e.g., collagens and fibronectin) contain integrin-binding motifs (such as RGD, GREGOR, and GLOGEN on collagen molecules). These protein domains interact with cell surface receptors, triggering downstream effects on cell fate and cellular processes [106,107]. Another mechanism that shows how ECM guidance affects cellular events is the accessibility and regulation of growth factor (GF) signaling. ECM proteins bind GFs and regulate their activity by providing a pericellular substrate for presenting GFs to specific cell receptors or by sequestering active and latent forms away from the cells for later utilization [105,108]. GFs bind to discrete domains and motifs of ECM proteins, and there is a remarkable specificity to these binding interactions. As the ECM changes during development, the same happens to an altered repertoire of ECM-associated GFs that influence cell phenotypes [105,109].

#### 4.2.2. ECM-Scaffold-Mediated Tissue Regeneration

An acellular scaffold retains constituent ECM proteins similar to the native organization as that in the source tissue. By repopulating cells in these acellular scaffolds, multiple studies have demonstrated the reformation of cell-enriched tissues and their therapeutic potential. Bone-derived ECM scaffolds combined with n-poly (e-caprolactone) nanofibers promoted the attachment, migration, proliferation, and osteogenic differentiation of rat MSCs [110]. Subsequent transplantation to a rat calvarial critical-size defect model mitigated the foreign body reaction and facilitated bone regeneration. Another study reported a composite of a calcium-silicate-enhanced small intestinal submucosa scaffold to be a 3D porous structure that improves the mechanical properties and promotes the osteogenic differentiation of human bone marrow MSCs in vivo [111]. An ECM scaffold derived from the submucosa of the small intestine promoted the repopulation of fibroblasts, blood vessels, and epithelium in periodontal tissues and peri-implant soft tissues, suggesting its potential use in soft tissue grafts [112]. Decellularized renal ECM containing poly (lactide-co-glycolide) magnesium hydroxide induced renal glomerular tissue regeneration after implantation to a partially nephrectomized mouse model [113]. Preclinical and human studies have also shown that ECM bioscaffolds can be used as an inductive substrate for tissue engineering applications in the gastrointestinal tract [114]. Although different studies have shown promising results, a long-term study needs to be performed to examine implant stability and host cell maintenance to ensure functional physiological homeostasis.

### 4.3. ECM Scaffolds in Drug Delivery and Tissue Engineering

#### 4.3.1. Scaffold-Mediated Drug Delivery

Scaffold-mediated drug delivery has two advantages: (i) it acts as a drug depot; (ii) it acts as wound dressing material, creating a physical barrier on the wound site [115]. Conventional drug delivery involves a simple diffusion-based release of drug molecules. For extended drug release, methods such as scaffold embedment [116] or nanomaterial coating on scaffolds [117] are being employed. For a precise control profile of drug release, the scaffold can be programmed as a hybrid scaffold containing natural and synthetic nanoparticles to achieve a stage-wise drug delivery. Furthermore, scaffolds can be modified to release drugs depending upon the physiologically relevant stimuli (e.g., pH). There have been different reports on the ECM-scaffold-mediated controlled release of drugs for wound healing. In a skin model, an injection of collagen hydrogel loaded with anti-fibrotic microRNA-29b to a rat epidermal wound resulted in improved remodeling of the skin’s ECM [118]. Other ECM proteins, such as collagen and hyaluronic acid, are also potential drug carriers for skin application. Tobramycin- and ciprofloxacin-loaded matrices were proven to have antibacterial effects for over 96 and 48 h, respectively [119]. The same study further showed that a tobramycin-loaded matrix containing bFGF significantly improved wound healing in a guinea pig skin wound model. More work has been conducted to evaluate drug release effects after the encapsulation of antibiotics, hydrophobic drugs, and inhibitors to ECM protein scaffolds [120,121]. Dreher et al. generated a temperature-responsive elastin-like peptide–doxorubicin conjugate, which was delivered to solid tumors for cancer therapy [122]. Using a similar elastin-like peptide model to conjugate antimicrobial peptides can be applied as an antimicrobial coating for a medical device [123].

Scaffold-mediated controlled drug release for wound healing can be as follows:

(i) Binary release of GFs: Choi et al. reported a synergistic effect in the wound healing of diabetic ulcers with bi-phasic release profiles of the loaded GFs. The growth factors (EGF and bFGF) were loaded into the scaffold matrix by encapsulating bFGF in the core and immobilizing EGF on the surface. At the wound site, approximately 30% of the encapsulated bFGF burst out in the first 12 h, whereas 2% of the immobilized EGF was released in 7 days. The initial burst of bFGF enhanced the proliferation of epidermal cells in the early wound repair, and the immobilized EGF provided a continuous stimulation of EGF receptors in keratinocytes, accelerating keratinocyte migration and proliferation during the entire wound healing process [124].

(ii) Sequential release of multiple GFs: a slower GF release pattern was reported in the layer-by-layer deposition of multiple GFs in nanofibers/scaffold layers, with independent control over the drug release rates in each layer [125].

(iii) On-demand drug release triggered by exogenous stimuli, such as enzymes, pH, etc. [126,127].

#### 4.3.2. Nanomaterials and ECM Scaffolds for Drug Delivery

Combining an ECM scaffold with nanoparticles (NPs) can promote targeted and controlled drug release and improve the bioavailability and biodistribution of therapeutic molecules [128,129,130]. An in vivo study on Parkinson’s disease drug delivery showed that the intracranial implantation of scaffolds embedded with dopamine-loaded cellulose acetate phthalate nanoparticles resulted in sustained drug delivery. The maximum dopamine entrapment efficiency was 63%, with the peak at day 3 when measured in rat cerebrospinal fluid and plasma [131]. This delivery system maintained an adequate level of dopamine for up to 30 days compared with the inherent dopamine levels. Fabricating an ECM scaffold with nanocarrier-encapsulated antibiotics, anti-viral drugs, or antifungal drugs can also be used to tackle secondary infections. Anti-viral drugs, such as dipeptide-acyclovir-based prodrugs encapsulated in poly (lactic-glycolic acid) (PLGA) nanoparticles, improved drug release kinetics [132]. In corneal therapy, idoxuridine packed in nanomaterial-based liposomes showed higher penetration through corneal tissues [133]. Higher retention of drugs on the corneal surface can also be achieved using NPs. Vichare et al. showed that chitosan NPs increased the retention time of the antifungal drug natamycin by 50% longer than the drug only [134]. Hence, this NP approach can be used with a corneal ECM scaffold to incorporate drug molecules. This could increase drug bioavailability and tissue penetrance. In a report by Chang et al., the loading of moxifloxacin and dexamethasone in nanostructural liposomal lipid carriers mixed with a collagen/gelatin/alginate substrate gave sustained drug release up to 12 h. This novel anti-inflammatory drug/scaffold formulation showed no cellular toxicity and instead promoted corneal epithelial cell proliferation and inhibited pathogen growth in vivo [135]. Improved ocular bioavailability of a modified version of a hyaluronic acid lipid–polymer hybrid NP was found with a higher drug permeability coefficient and drug retention, probably due to the surface-modified hyaluronic acid to improve the cellular uptake by receptor-mediated endocytosis [136]. In addition, a colloidal system with PLGA NPs was created for sparfloxacin ophthalmic delivery in corneas [137]. Compared with the marketed formulations with rapid clearance by systemic circulation, the new nanosuspension had significantly slower clearance and longer retention in corneal tissues and showed no irritation in a rabbit cornea model. This drug incorporation approach can be employed using a SMILE-derived lenticule scaffold to control transplantation-related complications, including infection and inflammation. However, the number of drugs being incorporated could be limited by the ultrathin nature of the lenticules. In addition, the efficient electrostatic bandpass of the ECM that could suppress the diffusive motions of the charged molecules could be different after FSL ablation [138,139]. Further studies are required to characterize these features and to understand if SMILE-derived lenticules are suitable candidates for controlled drug delivery.

Post-transplantation complications, such as corneal inflammation and neovascularization, reduce corneal transparency. Current treatments with topical corticosteroids, non-steroidal anti-inflammatory eye drops, and anti-VEGF-A antibodies work on suppressing the severity of these complications but with variable effectiveness. Gold NPs have been reported to have anti-inflammatory and anti-angiogenic properties [140,141]. Topically administered gold NPs inhibited experimental corneal neovascularization in mice [142]. In a rabbit corneal haze model, a topical application of the bone morphogenic protein 7 (BMP7) gene incorporated with polyethylimine-conjugated gold NPs significantly reduced corneal haze and inflammation, as revealed by a lower opacity level on the Fantes grading scale and a suppressed expression of αSMA, CD11b, and F4/80 [143]. This novel treatment modulated corneal wound healing and inhibited fibrosis by reducing TGFβ1-mediated pro-fibrotic Smad signaling. Hence, it is probable that ECM scaffolds combined with nanotechnology can bring about a significant change in drug delivery and tissue engineering. Further studies using SMILE-derived lenticule scaffolds with nano-modifications will be valuable for exploring its potential use in future corneal therapy and tissue engineering.

#### 4.3.3. Nanoparticles Incorporated in ECM Scaffolds

Natural hydrogels and ECM scaffolds provide the physicochemical and biological characteristics and properties suitable for cell growth. Further modulation of ECM scaffolds with nanomaterials could enable the matrix to be more conducive to cellular growth and survival, contributing to improved biocompatibility and integration [144,145,146]. Different studies have shown that incorporating bioactive NPs into the ECM enhanced the anti-inflammatory capacity of therapeutic hydrogels in a myocardial infarct model [147,148]. Combining gold NPs with an ECM to develop cardiac patches promoted MSC proliferation and the generation of cardiomyocytes [149]. Gold NPs distributed on the porous matrix surface could provide favorable conductivity and biological influences, such as intercellular electrical communications with cardiac cells. Electrically and biologically active gold NPs can increase the electrical behaviors of implanted ECMs for electrical coupling between cells and scaffolds to restore myocardial infarction. A cell culture showed that established patches enhanced biocompatibility and cardiomyocyte survival. In vivo transplantation results indicated that the gold NP-incorporated ECM patches decreased infarct size from 89% to 65%. In a musculoskeletal tissue model, an injection of combined gold NPs and ECM reduced tissue inflammation and improved cellularity and tissue remodeling due to a better release of growth factors. The incorporation of gold NPs was also shown to reduce reactive oxygen species levels [141].

### 4.4. Growth Factor Binding to ECM Scaffolds and Applications

#### ECM-Scaffold-Mediated Delivery of Growth Factors

Cytokines and growth factors (GFs) are soluble proteins released by cells that regulate various cellular processes and events in tissue development, repair, and regeneration [150,151]. Angiogenic factors, including vascular endothelial growth factor (VEGF), fibroblast growth factor (FGF), and placental growth factor (PlGF), are key players in wound healing [152,153]. Other cytokines, such as insulin-like growth factors (IGF), transforming growth factor-beta (TGF-β), bone morphogenetic proteins (BMP-2, BMP-4, and BMP-7), and platelet-derived growth factor-BB (PDGF-BB), regulate cell growth and tissue regeneration in various systems [154,155]. Despite their potential as therapeutic agents, the clinical use of cytokines has been limited due to their low stability, short effective half-life, and rapid inactivation and degradation by enzymes and proteases under physiological conditions [109,156]. For example, the half-life of basic FGF and VEGF is 3 and 50 min, respectively, after intravenous injection [157,158]. Hence, higher doses of GF are needed to achieve minimal effectiveness; however, this may lead to toxicity and adverse effects. An increased BMP-2 dose for lumbar spinal arthrodesis was reported to elevate the risk of carcinogenesis [159]. Owing to such restrictions, a controlled and localized release of cytokines and GFs is important to maximize their effectiveness and biologically relevant applicability, especially in tissue repair and regeneration [150,156].

ECM proteins contain specific motifs to bind multiple molecules, including GFs (Figure 4). Coupling the delivery of IGF and IGF-binding proteins to vitronectin in an ECM enhanced the cytokine functions in 2D and 3D culture models of human keratinocytes [160]. The cytokine level was maintained in wound fluid after topical application to a deep dermal wound in a porcine model. Alternatively, an engineered fibronectin substrate was shown to effectively bind various GFs (including VEGF-A, PDGF-BB, and BMP2) with potent synergistic signaling between α5β1 integrin and GF receptors, resulting in an enhanced tissue regenerative effect in a diabetic mouse model of chronic wounds, which was not observed using the fibrin delivery method [161]. Moreover, the ECM protein binding of GF can produce a long retention time and slow release at the target site. Compared with recombinant collagen, decellularized ECM hydrogel derived from a pericardial matrix effectively retained bFGF in ischemic myocardium and enhanced neovascularization with good anastomosis with existing vasculature [162]. With advances in the manipulation of ECM proteins and decellularized tissues across a range of biomaterials, and an increased ability to handle the presentation of GFs, this ECM biomimetic approach can improve the efficacy of tissue regeneration without reliance on supraphysiological doses of inductive biomacromolecules [163]. Furthermore, the cellular behavior and wound healing process are greatly impacted by these high-affinity ECM–GF associations [153,164]. For example, a modified hydrogel matrix coupled with fibronectin domains promoted the proliferation and migration of human dermal fibroblasts, which was not observed with RGD-tethered or unmodified hydrogel [165]. Furthermore, ECM interaction with GFs offers protection from degradation, hence controlling the bioactivity [166]. ECM proteins also bind to GF receptors, influencing downstream intracellular signaling [167]. Integrin binding to GFs can promote their activation, endocytosis, and recycling. Since the activity of GFs can control ECM biogenesis, their association creates a bidirectional relationship [168]. The ECM also acts as a reservoir for selected GFs, regulating their immobilization, controlled release, and bioactivity [109]. With all these facts and advantages, the affinity of the ECM with GF binding and release is being extensively studied to explore advanced delivery systems regulated in a spatio-temporal controlled manner and the future implications in the field of regenerative medicine [153,163].

The controlled release of GFs from the scaffold can complement the kinetics of physiological processes. Many factors can be independently released in scaffold-mediated delivery. Multiple preclinical studies have established the therapeutic potential of the ECM scaffold-mediated GF delivery [169,170,171]. Mesenchymal stromal cells seeded in a decellularized equine tendon scaffold containing TGFβ3 were promoted to undergo tenogenic differentiation, suggesting a potential for multiple GF delivery to treat tendon pathologies [172]. In wound healing processes involving different phases, from hemostasis/inflammation and fibrotic proliferation to ECM remodeling and scarring, a plethora of GFs influencing these phases direct the repair process. GF delivery integrated with ECM binding could modulate GF activity and signaling, even enhancing their efficacy when used at low doses [170]. This GF effect could influence the outcome of wound healing. Unfortunately, most of the studies on the biological actions of GF are from studies examining a single GF effect. The ECM scaffold/matrix could play a vital role in the GF-mediated wound healing process as follows:

(i) ECM–GF interactions activate GF receptor signaling. For example, ECM component heparin sulfate proteoglycans facilitate both the binding of FGF-2 to its receptor and the subsequent receptor dimerization, thereby promoting downstream FGFR signaling [166].

(ii) In addition to GF binding motifs on ECM, the regions or domains present in ECM proteins can directly bind to and induce the signaling of cell surface receptors. Swindle et al. showed that EGF-like repeats within the ECM molecule tenascin-C acted as a ligand for EGFR for activation [173].

(iii) Integrin-mediated indirect interactions between GF and ECM proteins influence various cellular processes where integrins act as a bridge between these molecules [167,168]. The binding of integrins to ECM initiates a downstream signaling cascade, which stimulates GF receptor signaling (Figure 4). In a vascular endothelial cell model, the cell surface αVβ3 integrins interacted with vitronectin, leading to VEGFR-2 signaling and enhancing the responsiveness to VEGF activity [174,175].

(iv) GFs controlled ECM production by increasing the synthesis of ECM components and/or modulating the production and activity of proteases that degrade and remodel ECM composition and organization. The pro-fibrotic TGFβ1 is known to stimulate collagen and various ECM protein production in fibroblasts and suppress ECM-degrading proteases [176].

## 5. Perspective on Translating Existing Knowledge of ECM-Based Scaffold Engineering to the SMILE-Derived Lenticule Scaffold for Potential Therapeutic Applications

With the native collagen-rich ECM composition, SMILE lenticules could be used to repair the host tissue defects after implantation. The lenticule scaffold can provide inductive niches and facilitate the recruitment and differentiation of host cells, thereby enhancing endogenous tissue regeneration [177]. The SMILE lenticule scaffold can be similarly explored for use in corneal tissue regeneration, especially in the stromal wound healing process [38,145].

Current studies of lenticule decellularization are performed at a laboratory scale mostly for research purposes. This process needs to be optimized and standardized with approved guidelines and international consensus to assess the quality of decellularized lenticule scaffolds before clinical use, especially for xenogenic or allogenic cases.

Decellularized lenticules can be repopulated with stromal cells and combined with regenerative cytokines, such as heparin-binding growth factor and TGFβ3, to accelerate wound healing and stromal regeneration. In a rabbit model investigating lenticule re-implantation, besides the restoration of corneal keratometric and topographic indices to near preoperative values, host stromal cells were observed to repopulate at the lenticule borders, which could be more intensified in the long run [178]. This provides proof-of-concept cell restoration on decellularized lenticules after implantation. More recently, our group also reported re-innervation to the decellularized lenticules using a chick dorsal root ganglion model [91]. Successful recellularization and re-innervation will be beneficial for the integration of implanted lenticules with the surrounding recipient tissue. A regenerative effect could be achieved by long-term quiescent stromal cell infiltration and repopulation, which can promote stromal collagen turnover and tissue remodeling.

Moreover, SMILE lenticules could facilitate regenerative cytokine delivery. The roles of different GFs (detailed in previous sections), including the EGF family, KGF, HGF, IGF, insulin, and TGF-β, have been established in corneal wound healing [179,180,181]. A study by Seif-Naraghi et al. showed that injectable decellularized ECM-derived hydrogel provided a platform for the enhanced retention and delivery of HGF. The hydrogels retained native sulfated GAG that had GF-binding domains, and this could provide an excellent delivery platform for GF with improved stability and activity [162]. The corneal stromal ECM is a rich source of sulfated GAGs; hence the SMILE lenticule scaffold could be developed as a potential carrier of regenerative cytokines. This would provide targeted delivery and the controlled release of GFs in the recipient tissues.

Some limitations of tissue-engineered corneal transplants are suboptimal integration and the risk of tissue rejection, especially in the case of treating acutely damaged and highly inflamed tissues [37,182]. Poor interactions between the graft and host tissue could influence multiple molecular events, including healing, inflammation, ECM remodeling, and cell death at the graft host site [16,129,183]. Corneal scaffold fabrication with nanomaterials has the potential to improve the physicochemical properties that enhance graft integration in the host tissue [129]. They can also be fabricated with NPs, such as silver or gold NPs, to modulate wound healing and for cell repopulation. In addition, NPs can be used to manage post-implantation complications, such as inflammation, secondary infections, and neovascularization in the host area. Various nanocarrier-based therapeutic agents can be used in the scaffold fabrication phase [129,184]. With such scenarios as these, the use of lenticule scaffolds incorporated with gold NPs for stromal implantation could enhance stromal cell viability, manage inflammation by regulating lymphocytic and platelet activation after injury, and potentially have an impact on nerve growth and repair in tissue regeneration.

## 6. Conclusions

Over the last decade, considerable progress has been made in the research of ECM-scaffold-mediated tissue regeneration. In the treatment of corneal diseases, cell-based and tissue-engineered therapies are being explored in preclinical studies, and some promising studies have even reached the clinical stage. These therapies can be considered potential alternatives to donor corneal tissues, which have a global issue of limited availability. With the growing popularity of SMILE surgery for refractive corrections, the extracted lenticules with native collagen-rich stromal organization and high mechanical strength and transparency are worth being recycled or upcycled for therapeutic uses instead of being disposed of as medical waste. The successful development of decellularization protocols has paved the way for the generation of high-quality transplantation-worthy stromal ECM scaffolds that are appropriate for multiple avenues of fundamental and translational regenerative research. The advancement of cell repopulation on these ECM scaffolds and the modifications with GFs and nanoparticles highlight the potential uses as tools for tissue regeneration and in the development of drug and GF delivery systems. Further research into SMILE-derived lenticule scaffold-based regenerative therapeutics along these lines will benefit not only corneas and ocular surfaces but also other tissue organs, such as skin and tendons.

## Figures and Tables

**Figure 1 ijms-23-07967-f001:**
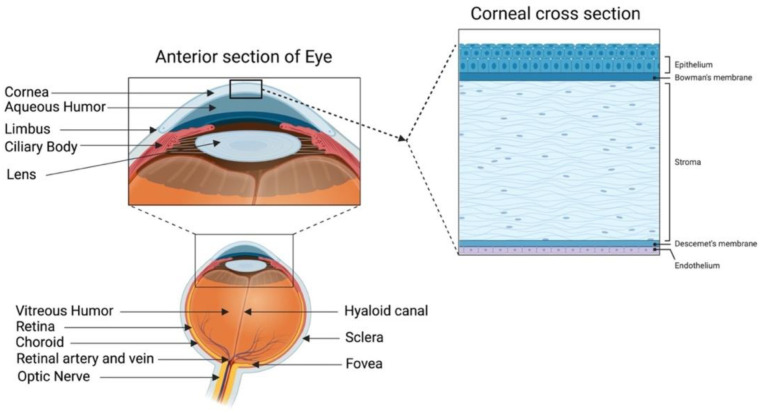
Human cornea: anatomy and structure. In the anterior segment, the cornea is highlighted in relation to the rest of the eye. A schematic representation of the structure and composition of the cornea is presented in the corneal cross-section. It consists of 3 cellular layers (epithelium, stroma, and endothelium) and 2 basement membranes (Bowman’s layer and Descemet’s membrane) (figure created using BioRender.com, BioRender, Toronto, ON, Canada).

**Figure 2 ijms-23-07967-f002:**
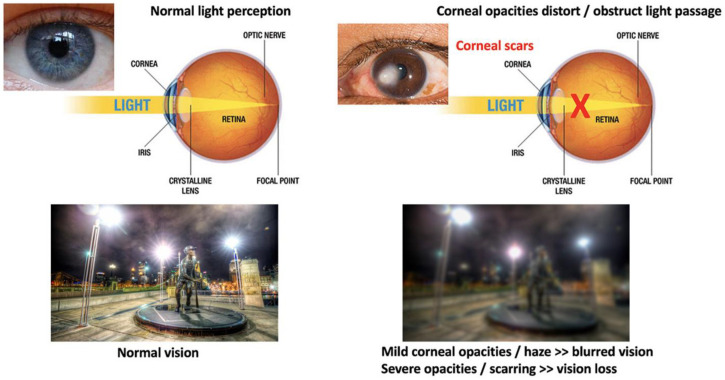
Comparison of vision between clear and opaque corneas and conventional corneal grafting to replace scarred corneas. Normal vision with unblocked light passage through a healthy and clear cornea, leading to clear and sharp visual acuity. The presence of corneal scarring/opacities blocks light passages, resulting in vision loss (corneal blindness).

**Figure 3 ijms-23-07967-f003:**
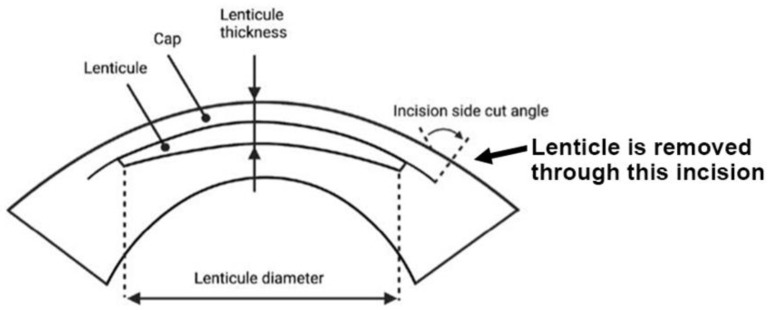
SMILE, a lenticule-based procedure. Side view of the lenticule profile and lenticule removal via the small incision (bold arrow) (Figure created using BioRender.com, Toronto, ON, Canada).

**Figure 4 ijms-23-07967-f004:**
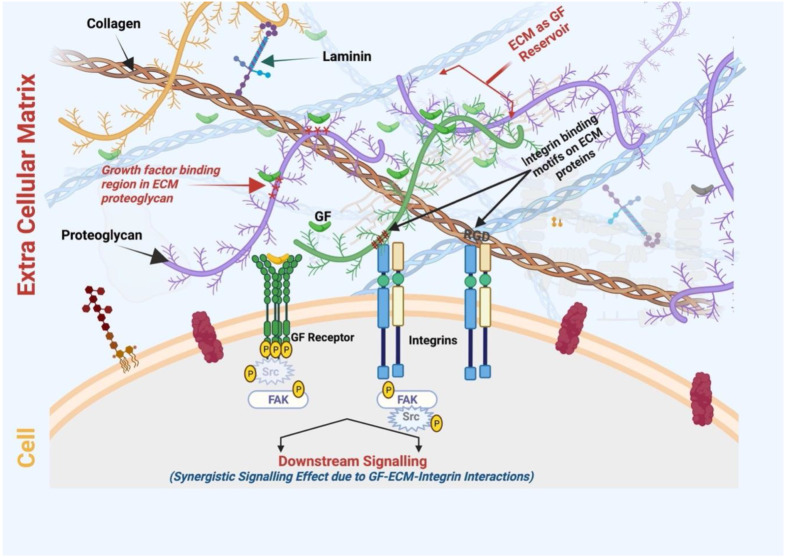
ECM act as GF reservoirs where proteoglycans contain GF binding domain/regions. Various ECM proteins such as collagen and fibronectin contain integrin-binding motifs (for example, RGD or the Arg-Gly-Asp motif) that bind to the cell surface integrin receptor. Therefore, ECM-scaffold-mediated GF delivery can simultaneously induce the signaling cascades of integrins and growth factor receptors, resulting in increased and prolonged GF signaling. SMILE-derived ECM-scaffold-mediated GF delivery can be exploited for its synergistic wound healing/regenerative therapies at a lower dose of GF (Figure created using BioRender.com, BioRender, Toronto, ON, Canada).

## Data Availability

All data are included in the text.

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
