# Peer review of "Human SMILE-Derived Stromal Lenticule Scaffold for Regenerative Therapy: Review and Perspectives"

_ijms, 2022, doi:10.3390/ijms23147967_

Round 1

Author Response

This is an interesting review paper regarding the use of human refractive surgery-derived corneal scaffolds for regenerative therapy purposes. The authors have identified that cornea transplantation is the only treatment option with high success; however, the restriction of donors requires the development of alternative treatment options. More specifically, this review focuses on the use of ECM-scaffold based engineering and on the tissue derived from the Small Incision Lenticule Extraction (SMILE).

In general, the authors explain the background, and discuss the gaps and perspectives, but some parts are discussed very superficially without being supported with enough literature. Therefore, it might be more appropriate to remove some of them or pull them together and describe them in more detailed.

  Reply – We thank the reviewer for the effort and constructive comments. Changes were red highlighted in the revised version keeping the reviewer's suggestions in mind.

  1. The title should include the “SMILE” since the review focuses more in this approach

  Reply – Thank you for the suggestion. The title has been revised as “Human SMILE-Derived Stromal Lenticule Scaffold for Regenerative Therapy: Review and Perspectives”.

  1. Section 1.3: Elaborate more about the molecular mechanisms of wound healing. For example, CSK maintain cornea homeostasis by regulating ECM turnover not only by depositing ECM. Additionally, upon injury the CSK subjected to phenotypic alterations to restore tissue integrity, more information about it should be added.

  Reply – Thank you for the suggestion. In the revised manuscript, we have added more details about the cellular and molecular mechanisms of stromal wound healing and corresponding references (page 3, third paragraph).

  1. Lines 126-129: add more specific references and more research papers for the topics of the epithelial, stromal and endothelial cells

  Reply Thank you. More details for each topic have been added on page 3 of the revised manuscript (2nd paragraph).  

  1. Lines 130-131: “Transplanting an engineered product...” may be does not fit in the cell-based therapies section

  Reply Thank you. It was removed in the revised manuscript.

  1. Section 2.1.1 This section either should be moved as a conclusion of the previous paragraph or explained in more detail.

  Reply Thank you. In the revised manuscript (on page 4, 2nd paragraph), we removed 2.1.1. and merged both paragraphs in the same section. More details about the limitations of cell-based therapy have been added, to emphasize the importance of scaffold-based strategy.

  1. 6. Line 158: Which limitations?

  Reply Thank you. The limitations are those that are mentioned in the preceding paragraph. We have clearly stated this in the revised manuscript (on page 4, line 167-172).

  1. Line 164-165: It should be given more specific examples for connective tissues that share more similarities compared to muscle

  Reply Thank you for your suggestion. This point has been elaborated on along with specific references (on page 6, first paragraph). We have given examples of the corneal stroma, tendons, cartilage, bone, and basal epithelium. 

  1. Section 2.2.1 Explained poorly, better to remove it since it does not add much to the whole concept.

  Reply Thank you. The part of synthetic materials was removed.

  1. Line 207: Explain briefly the pros and cons of the protocols, give specific examples

  Reply Thank you. Most details of decellularization protocols, efficiency, and outcomes have been broadly reviewed in several papers, which are listed in this revision. In addition, we have added several limitations based on our experience of corneal tissue decellularization. These points are revised on pages 9 to 10.

  1. Line 217: Why the lenticule is extracted? Explain with more detail

  Reply Lenticule extraction is the proper term to describe the removal of lenticule from the corneal stroma. That is why SMILE refers to SMall Incision Lenticule Extraction. In order not to cause any misunderstanding, we revised it to “lenticule is removed” (on page 6, 2nd paragraph).

  1. Line 225: What is LASIK? Explain more about LASIK to be more obvious why it is brought up as a comparison example or remove the LASIK mention.

  ReplyThank you for the suggestion. The section on LASIK was removed as this review is focused on SMILE-derived lenticules.

  1. Line 239-241: Explain better, why and how SMILE-derived stromal lenticules can be useful products. Connect it with the concept of the already mentioned sections such as scaffold bioengineering, decellularisation etc.

  Reply  Thank you for the comments. The reported uses of stromal lenticules in pre-clinical and clinical studies have been added on page 7. And Table 1 illustrates the details of the clinical study using lenticule implantation. A perspective using SMILE lenticule scaffold for the delivery of cells, drugs, growth factors, and nano-modification is discussed in section 5.

  1. Whole Section 4: Different topics are discussed but not in detail. This whole section does not fit well with the rest of the review and therefore, should be removed.

  Reply Thank you for the comment. The part of organoid generation on decellularized scaffolds was removed whereas recellularization was revised with more details showing the ECM scaffold can be a promising vehicle to support cell transplantation with the advantage of precise location and niche supporting cell growth and adhesion. This revision is on page 10, 4th paragraph.

  1. Sections 5.2, 5.3, 54: Too small sections that have been approached superficially. Can be pulled together in one section and discussed more appropriately.

  Reply Thank you for the suggestion. The 3 subsections were combined in section 5 on page 16.

  1. Sections 6.1, 6.2: The same as above.

  Reply Thank you. The limitations and potential obstacles were merged into Section 5, the last paragraph.

Author Response

  1. The authors propose an innovative approach in harvesting the excised lenticules after SMILE for use in corneal transplantation. Although globally this seems to be a rarer procedure, the authors state that it is gaining popularity. Furthermore, an otherwise healthy supply of corneal stromal tissue from healthy, living donors who can offer their consent could provide valuable donor material. Corneal transplantation is a much-needed procedure globally as the review points out and unfortunately due to a number of reasons there is a gap between the number of transplantations needed and the number performed. Donor supply is just one of the factors preventing transplantations from occurring and even within that there are a number of factors determining the availability of tissue; Some of which (presumably) could be overcome by SMILE-derived donor material (patients may be more likely to consent to the use of tissue for example) and some of which could not - the incidence of corneal opacities (and therefore demand) is likely to be highest in countries where refractive surgeries occur least for example. The review does not deeply dissect the logistics of weather this procedure could meaningfully change the gap between need and transplantation. Instead it predominantly covers the broadly surrounding biology of (already well reviewed) regenerative medicine in relation to corneal disease. Overall, it overly-optimistically reviews evidence in support of SMILE-derived lenticule applications and perhaps fails to be critical in its potential/probable limitations. At points the review draws lessons from other tissues of the body, perhaps the authors should make it clear at these points when they have diverged from discussing the cornea specifically and maybe consider the relevance of comparing meniscus, tendon or other tissues against the particular constraints associated with the cornea (and in doing so making the review more concise).

  Reply Thank you for the great effort in reviewing our manuscript and the constructive and critical comments. Compared to LASIK, SMILE is definitely a relatively new refractive procedure designed to treat a multitude of refractive errors, such as myopia, hyperopia, presbyopia, and astigmatism. It is reported to achieve more stable and predictable effects than LASIK and with excellent postoperative outcomes, such as faster recovery, less incidence and severity of post-op dry eye, faster reinnervation, and potential biomechanical advantage. Unlike LASIK, complications arising during SMILE have been reported very infrequently, supporting the reported safety and predictability of the procedure. Hence, SMILE will be an upcoming option for more patients undergoing refractive surgeries. In the revision, we have stated these on page 6. We also added the recent report by Schallhorn et al. in Cataract & Refractive Surgery Today’s, 2021 about the significantly increased volume of SMILE procedures performed worldwide. More than 3.5 million SMILE procedures have been performed globally and more than 70 countries have been approved to perform SMILE.

Since SMILE is an up and rising technique in the recent decade, the SMILE-derived lenticules, instead of being discarded as medical waste, can be a great resource of native collagen-rich and mechanically strong biomaterial that is worth to be explored for potential uses. Because very limited studies have reported on the potential uses of these lenticules, like as a cell carrier or vehicle for drug delivery or other molecules, the aim of this review and perspective is to exploit the existing knowledge based on ECM scaffolds derived from other tissue sources. This will help us or other researchers identify the areas of interest and the essential questions to be investigated. We have stated this on page 10, 3rd paragraph. In the revised version, we have elaborated in more detail how different studies on ECM scaffolds be applicable for SMILE-derived lenticules. All changes were red highlighted for easy tracking. 

  1. It is unclear why keratocytes have been abbreviated to CSK - amend for clarity.

  Reply Thank you. CSK is the abbreviation of corneal stromal keratocytes, and has been commonly used in publications. In this revision, we removed the abbreviation and used “keratocytes” throughout the manuscript.

  1. Section 2.2: Entitled scaffold-based bioengineering is actually largely about cell/scaffold combinations. Whilst this is a well-known approach to corneal tissue engineering the title (particularly with respect to the previous section on cell only transplantation) makes it seem as though it would be scaffold only (which is also a well-established methodology).

  Reply Thank you. We apologize for the incorrect sub-title and misunderstanding. It should be “Scaffold-based cell delivery strategies” (on page 5, first paragraph).

  1. Section 3: Stromal lenticules from SMILE. Whilst I believe the idea has merit, in order for me as a reader, to understand the power of it I feel the authors need to explain more about the practicalities here. The authors describe a specific procedure in laser refractive surgery that (outside of China) is performed around 45,000 times a year globally. Is this a lot? Is it performed in locations where tissue can be aseptically prepared for shipping and storage to a centralised facility for preparation and distribution nationally? Globally? Of the countries where SMILE is more common, what are the limiting factors preventing donation or transplantation? Do primary sources explain how many lenticules pass/fail requirements for donation? Are patients likely to donate this tissue? If all tissue from SMILE could be harvested, how much of an impact could this have on donor supplies?

  Reply Thank you. Based on a recent study, we have updated that over 3.5 million SMILE procedures were performed globally in the last 10 years since its commencement in Sep 2011, accounting for an average of 350,000 cases per year. This point has been added on page 6 (paragraphs 2 – 4). The clinical SMILE procedures are regulated and approved by FDA or equivalent government agencies. Hence, tissues are collected with consent and under aseptic conditions. The ultrathin nature of lenticule also allows the use of multiple aseptic washes to safeguard tissue sterility. Different studies have reported lenticle processing, customization (reshaping and thinning, collagen crosslinking and decellularization), shipping, storage, and banking, including  

  1. Yam et al., 2016 Sci Rep “Decellularization of human stromal refractive lenticules for corneal tissue engineering”
  2. Liu et al., 2017 Mol Vis “Corneal lenticule storage before reimplantation”
  3. Riau et al., 2021 Tissue Eng Part A “Experiment-Based Validation of Corneal Lenticule Banking in a Health Authority-Licensed Facility”
  4. Yam et al., 2022 J Adv Res “Effect of corneal stromal lenticule customization on neurite distribution and excitatory property”
  5. Damgaard et al., 2018 Invest Ophthalmol Vis Sci “Reshaping and Customization of SMILE-Derived Biological Lenticules for Intrastromal Implantation”

The patient factors limiting SMILE include low myopia, high astigmatism, and difficult orbital anatomy. A patient’s pre-operative assessment is important for the success of the surgery. Patients with abnormal corneal topography and forme fruste keratoconus must be excluded. If the percentage of tissue altered (calculated as lenticule thickness + cap thickness)/central corneal thickness is more than 40%, the patient has an increased risk of ectasia. Moreover, patients with deep-set eyes, with small palpebral fissures, or being anxious or uncooperative are prone to affect the SMILE procedures that involve suction loss and poor docking of femtosecond laser probe.

Are patients likely to donate this tissue?

With SMILE gaining worldwide acceptance among refractive surgeons and different modifications of surgical techniques have been described to ease the process of lenticule extraction and minimize complications, the willingness of patients to undergo SMILE procedures is expected to increase. In addition, the established protocols of lenticule storage and banking have improved the patients’ acceptance to undergo SMILE as the autologous lenticles extracted from myopic correction can be banked for future use in the treatment of presbyopia, hyperopia, aphakia, and corneal perforation, once the reimplantation techniques become mature.

If all tissue from SMILE could be harvested, how much of an impact could this have on donor supplies?

This point needs further studies to clarify.

  1. Section 3.1: I feel this section is key to the novelty of the review and would like to see more discussion on these primary data publications, their pros/cons/next steps etc. However I feel their supposition on rejection at the end of this section lacks credibility.

  Reply Thank you for your appreciation. The reuse of SMILE lenticules is in early stage and various works is still underway to investigate how to achieve better utilization of these lenticules. Hence, the available publication is limited. In the revised section we highlighted the current obstacles in reusing lenticules for clinics (section 3.4 on page 9).

  1. Section 3.3: although plausible, autologous iPSC-derived autologous transplantation is very difficult and currently impractical, particularly when limbal stem cell transplants (both autologous and allogeneic) are widely performed. Furthermore, autologous cells on an allogeneic scaffold (albeit decellularized) are not guaranteed to eliminate graft rejection but could substantially lower the risk.

  ReplyThank you for the comment. Though more thorough studies are needed to demonstrate the safety and long-term efficacy of autologous iPSC treatments, we believe this strategy could be plausible for treating patients with bilateral persistent epithelial defects due to total limbal stem cell deficiency in both eyes. This part has been revised on page 9 (3rd paragraph). We also agree that the risk of immune response and tissue rejection may not be totally eradicated by decellularization. However, the risk will be much reduced and can be readily managed by other standard treatments, like topical steroids.

  1. Section 4.1: Recellularisation of tissues is a difficult process and while this section provides some evidence superficially to cover the literature to date, it does not critically evaluate the research. How plausible is it to expect a decellularized stroma to remain mechanically and optically relevant and could such a scaffold be easily recellularised? Do the processes of stromal cell migration, proliferation and culture change their phenotype and prevent them becoming quiescent cells capable of maintaining transparency? There are no fixed “good manufacturing products [sic]” guidelines – regulators of each country will approve a therapy based on the process presented to them - the authors may want to reconsider this sentence.

  ReplyThank you. We agree that the lenticule modifications, like de- and re-cellularization, are still at an early stage of research. In particular, the efficiency of re-cellularization is expected to be different between primary and immortalized stromal cells, which we have stated on page 11. Among primary cells obtained from the same type of stromal tissues, their feature differences, like the extent of cell migration and growth, could also influence the efficacy of cell binding to the scaffold. Nevertheless, successful stromal cell repopulation in the lenticules has been described in a rabbit re-implantation model (Angunawela et al., 2012 Invest Ophthalmol Vis Sci). Recently, we further demonstrated the reinnervation of the decellularized lenticules using a chick dorsal root ganglion model (Yam et al., 2022 J Adv Res). Thus, lenticule recellularization is highly plausible but it needs more extensive investigations to standardize the protocols and improve their efficiency.

Decellularized lenticules are demonstrated to have high optical clarity. In our earlier study, the mild detergent treatment of human stromal lenticules with 0.1% SDS achieved efficient decellularization and denucleation while the resulting scaffolds had similar clarity, demonstrated by the spectral transmittance (380 to 780 nm range), as the naïve lenticules of an equivalent thickness (Yam et al., 2016 Sci Rep). While the mechanical properties, like the tensile strain and stiffness, could be affected to some extent by decellularization, which deprives cellular materials of the stromal matrix, the tissue strength can be improved by collagen crosslinking with the UVA-riboflavin method. Our study in Yam et al. 2022 J Adv Res has illustrated this mechanical restoration of stromal lenticles.

GMP guidelines – we agree to the reviewer’s comment and have removed this short paragraph. 

  1. Section 4.2 (including subsections): I feel this section is tenuous at best. I’m not sure which organoids the authors believe can be grown in or on lenticules? Most organoids form in 3D cultures and those which use ECM substrates, tend to use thin, liquid layers to coat tissue culture surfaces. Are the authors suggesting there is a way to process the lenticules for corneal specific differentiation? This section also seems to repeat many previously discussed points, which is unnecessary and requires consolidation/streamlining. As mentioned in the general comments this section struggles with focus and specificity.

  Reply Thank you for the comment. The part of organoid generation on decellularized scaffolds was removed.

  1. Section 5 similarly seems to be a review of points raised in sections 1-4

  ReplyThank you. Section 5 states the perspectives to translate the existing knowledge obtained from other ECM scaffolds (stated in Section 4) to the SMILE lenticule scaffolds. We anticipate these would shed light on exploring the novel potential of SMILE-derived lenticules in future applications.